# Multimodal Treatment of Pleural Mesothelioma with Cytoreductive Surgery and Hyperthermic Intrathoracic Chemotherapy: Impact of Additive Chemotherapy

**DOI:** 10.3390/cancers16081587

**Published:** 2024-04-20

**Authors:** Laura V. Klotz, Julia Zimmermann, Karolina Müller, Julia Kovács, Mohamed Hassan, Michael Koller, Severin Schmid, Gunnar Huppertz, Till Markowiak, Bernward Passlick, Hans-Stefan Hofmann, Hauke Winter, Rudolf A. Hatz, Martin E. Eichhorn, Michael Ried

**Affiliations:** 1Department of Thoracic Surgery, Thoraxklinik, University Hospital Heidelberg, 69126 Heidelberg, Germany; 2German Center for Lung Research (DZL), 69120 Heidelberg, Germany; 3Division of Thoracic Surgery, Ludwig-Maximilians-University Munich, Asklepios Lung Clinic Gauting, 82131 Gauting, Germany; julia.zimmermann@med.uni-muenchen.de (J.Z.); julia.kovacs@med.uni-muenchen.de (J.K.); 4Center for Clinical Studies, University Hospital Regensburg, 93053 Regensburg, Germany; 5Department of Thoracic Surgery, University Hospital Freiburg, 79106 Freiburg im Breisgau, Germany; 6Department of Thoracic Surgery, University Hospital Regensburg, 93053 Regensburg, Germany; 7Department of Thoracic Surgery, Barmherzige Brüder Hospital Regensburg, 93049 Regensburg, Germany

**Keywords:** pleural mesothelioma, chemoperfusion, cytoreductive surgery, cisplatin, thoracic surgery, multimodal, decortication

## Abstract

**Simple Summary:**

Cytoreductive surgery combined with intraoperative hyperthermic chemoperfusion (HITOC) within a multimodal treatment approach has a positive impact on the survival of highly selected patients with epithelioid pleural mesothelioma. The addition of chemotherapy significantly affects the interval to tumor recurrence.

**Abstract:**

Cytoreductive surgery (CRS) combined with hyperthermic intrathoracic chemoperfusion (HITOC) is a promising treatment strategy for pleural mesothelioma (PM). The aim of this study was to evaluate the impacts of this multimodal approach in combination with systemic treatment on disease-free survival (DFS) and overall survival (OS). In this retrospective multicenter study, clinical data from patients after CRS and HITOC for PM at four high-volume thoracic surgery departments in Germany were analyzed. A total of 260 patients with MPM (220 epithelioid, 40 non-epithelioid) underwent CRS and HITOC as part of a multimodal treatment approach. HITOC was administered with cisplatin alone (58.5%) or cisplatin and doxorubicin (41.5%). In addition, 52.1% of patients received neoadjuvant and/or adjuvant chemotherapy. The median follow-up was 48 months (IQR = 38 to 58 months). In-hospital mortality was 3.5%. Both the resection status (macroscopic complete vs. incomplete resection) and histologic subtype (epithelioid vs. non-epithelioid) had significant impacts on DFS and OS. In addition, adjuvant chemotherapy (neoadjuvant/adjuvant) significantly increased DFS (*p* = 0.003). CRS and HITOC within a multimodal treatment approach had positive impacts on the survival of patients with epithelioid PM after macroscopic complete resection. The addition of chemotherapy significantly prolonged the time to tumor recurrence or progression.

## 1. Introduction

A rare and highly aggressive malignancy, malignant pleural mesothelioma (PM), is located in the pleural cavity and is primarily associated with asbestos exposure [1]. The widespread use of asbestos over the past decades coupled with a long latency period to disease onset are responsible for a continuously relevant incidence of PM [2]. Although immunotherapy has recently been introduced as a first-line treatment, breakthrough improvements in overall survival are still lacking [3,4]. In addition, there is a lack of data on multimodal treatment concepts that include cytoreductive surgery combined with intracavitary therapies and chemotherapy.

Cytoreductive surgery (CRS), defined as pleurectomy/decortication (PD) or extended pleurectomy/decortication (ePD) with resection of diaphragmatic or pericardial structures or extrapleural pneumonectomy (EPP), within a multimodal treatment approach allows the prolongation of overall survival. However, local tumor recurrence often occurs within a short period of time due to microscopic tumor deposits remaining in the chest, as there is no safety margin and, at best, macroscopic complete resection (MCR) can be achieved by CRS [5,6]. Recent studies have demonstrated a clear advantage of lung-sparing cytoreduction within a multimodal treatment approach over EPP in terms of overall survival [7]. The introduction of hyperthermic intrathoracic chemotherapy (HITOC) after CRS aims to improve local tumor control and prolong recurrence-free and overall survival [8]. In addition, the effect of adjuvant chemotherapy within a multimodal treatment approach may have an impact on residual tumor cells.

Due to the rarity of PM and delayed diagnosis at advanced stages, there is still no standardized multimodal treatment approach for localized disease [9,10]. Few studies have been published on multimodal treatment with CRS, HITOC, and neoadjuvant/adjuvant chemotherapy in patients with PM [7,8,11,12,13]. We postulated the study hypothesis that patients with PM may particularly benefit from complete multimodal therapy including tumor resection and intracavitary and systemic chemotherapy.

The aim of this retrospective multicenter study was to evaluate the impacts of CRS and HITOC within a multimodal treatment concept for patients with PM on disease-free survival (DFS) and overall survival (OS) and, in particular, to assess additional chemotherapy and relevant prognostic factors.

## 2. Patients and Methods

### 2.1. Patient Selection

In this retrospective multicenter study, clinical outcomes of 260 patients with a primary diagnosis of epithelioid or non-epithelioid PM from the German HITOC study were analyzed (Figure 1). Between 2008 and 2019, four high-volume departments of thoracic surgery in Germany collected clinical data on pre-, peri-, and postoperative variables according to a common protocol [14]. The study was funded by the German Research Association (DFG, RI2905/3-1). The trial was registered in the German Registry of Clinical Studies (DRKS-ID: DRKS00015012). The approval of the Ethics Committee of the University of Regensburg (No. 18-1119-104) and of the ethics committees of the respective participating centers were obtained. All patients underwent elective CRS and HITOC within one surgery. The surgical procedures and HITOC have been conducted as described before [12,15]. The existing database was supplemented by the additional collection of 64 clinical and follow-up data, which were updated until November 2021. One patient from the entire study population (n = 261) was excluded because he was initially treated for local recurrence of PM.

### 2.2. Definition of Variables

The histological subtype was categorized in epithelioid vs. non-epithelioid (biphasic and sarcomatoid). UICC stage was categorized as stage I ≥ II according to the 8th UICC tumor classification. Additive chemotherapy was defined as either induction chemotherapy and/or adjuvant chemotherapy. The beginning of primary therapy was defined as the date of surgery if no induction chemotherapy was applied. If induction chemotherapy was applied, beginning of primary therapy was set as the date of the first application of induction chemotherapy. HITOC was performed for 60–90 min at 42 °C with a flow rate of approximately 1000 mL per minute, depending on the standard operating procedure of the participating centers. The concentration for intrathoracic perfusion with cisplatin was categorized as a low (≤125 mg/m^2^ BSA = body surface area) vs. high (>125 mg/m^2^ BSA) dose. Overall survival (OS) was defined as the time from the beginning of the primary therapy until death from any cause. Disease-free survival (DFS) was defined as the time from the beginning of the primary therapy until the first objective tumor recurrence/progression or death from any cause, which ever occurred first. Recurrence or progression were defined as documented intrathoracic ipsilateral and/or contralateral tumor detection and distant metastases by cytology/histology and/or imaging. Patients received additive chemotherapy or follow-up following CRS and HITOC depending on the tumor stage, resection status, and the recommendation of the interdisciplinary tumor board at each of the four study centers.

Primary objective of the study was to analyze the OS rates of patients with PM who underwent CRS and HITOC within a multimodal treatment regimen. Secondary endpoints included DFS and the identification of prognostic factors.

### 2.3. Statistical Analysis

Descriptive analyses were performed for demographic, treatment, postoperative, and follow-up data using frequencies (n), percentages (%), means (m), standard deviations (SDs), medians (med), and interquartile ranges (IQRs). Median follow-up time was calculated using the reverse Kaplan–Meier method. OS and DFS were analyzed by the Kaplan–Meier estimator for the total patient cohort. Moreover, univariable analyses (log-rank tests) were used to compare OS and DFS between the cisplatin dosage, chemotherapeutical agent, resection status, histological subtype, UICC (Union Internationale Contre le Cancer Classification) stage, postoperative acute kidney injury, and additive chemotherapy. The estimates for the probability of surviving were presented graphically in Kaplan–Meier survival curves. Additionally, multivariable Cox proportional hazard regression models (enter method) were calculated. For the endpoints OS and DFS, the abovementioned clinical parameters (cisplatin dosage, chemotherapeutic agent, resection status, histological subtype, UICC stage, postoperative acute kidney injury, and additive chemotherapy), as well as the confounding variable time between the initial diagnosis and beginning of primary therapy, were included. Hazard ratios (HRs) with corresponding 95% CIs are presented. All statistical analyses were conducted using the software package SPSS (version 26 or higher). The significance level was set at *p*_two-sided_ ≤ 0.050.

## 3. Results

The HITOC database contains a total of 350 patients. In total, 260 patients met the prespecified inclusion and exclusion criteria and were included in the present analyses. The mean age of the patient cohort was 65.5 ± 9.0 years, and 81.2% were male. Good performance status with ECOG (Performance Status Scale of the Eastern Cooperative Oncology Group) ≤ 1 was documented in 95.4% of the patients. Regarding tumor histology, 84.6% had the epithelioid tumor subtype, 13.5% had the biphasic subtype, and 1.9% had the sarcomatoid tumor subtype. According to tumor stage, 48.1% had stage I disease, 15.0% had stage II disease, 34.6% had stage III disease, and 2.3% had stage IV disease. Detailed patient characteristics are shown in Table 1.

### 3.1. Surgical Approach

CRS by lung-sparing PD was performed in 60 patients (23.1%), while ePD was performed in 193 patients (74.2%), including partial resection of the diaphragm in 173 patients (66.5%) and partial resection of the pericardium in 118 patients (45.4%). Partial resection of the chest wall was performed in 25 patients (9.6%) due to local tumor infiltration. Only 2.7% of patients underwent EPP. Macroscopic complete resection (MCR) was achieved in 222 patients (85.4%) and only 38 patients (14.6%) had macroscopic incomplete resection (R2) due to tumor infiltration into non-resectable structures. MCR was defined as the resection of all of the visible tumor with a residual tumor volume of less than 1 cm^3^, as defined by Sugarbaker and colleagues [6,16]. The median blood loss during surgery was 800 mL (IQR = 500, 1200). In total, 145 (55.8%) patients received a transfusion of erythrocytes, platelets, or fresh frozen plasma during surgery. Detailed characteristics are shown in Table 2.

Intrathoracic chemotherapy (HITOC) consisted of cisplatin as a single dose (58.5%) or in combination with doxorubicin (41.5%) and was administered with low-dose cisplatin (≤125 mg/m^2^ BSA; 73.2%) or high-dose cisplatin (>125 mg/m^2^ BSA; 26.8%) according to the capabilities of the participating center.

### 3.2. Postoperative Data

Detailed postoperative data are shown in Table 3. Total postoperative complications were documented in 142 patients (54.5%), while Clavien–Dindo grade I and II complications were documented in 24.6%. Complications requiring surgical intervention (Clavien–Dindo IIIb) occurred in 37 patients (14.2%) due to prolonged air leak and coagulo- or chylothorax. Grade IV complications occurred in 1.6% and in-hospital mortality was 3.5%. The median intensive care unit (ICU) stay was 2 days (IQR = 1 to 4 days), while the median hospital stay was 20 days (IQR = 15 to 29 days).

### 3.3. Multimodal Treatment and Follow-Up

A total of 136 patients (52.3%) received additive chemotherapy. Within this patient cohort, 50 patients received neoadjuvant chemotherapy, while 73 patients received adjuvant chemotherapy, and 13 patients received both neoadjuvant and adjuvant chemotherapy. Median follow-up, defined as the time from the start of primary therapy to the end of follow-up, was 48 months (95% CI = 38.4, 57.6). During follow-up, 157 patients (80.5%) had locoregional tumor recurrence (in the case of MCR) or tumor progression (when macroscopic incomplete resection (R2) was documented). Distant metastases with or without locoregional recurrence or progression occurred in 32 patients (14.5%). Tumor recurrence or progression was treated with chemotherapy in 117 patients (60.0%) and radiotherapy in 30 patients (15.4%). Details of additive treatment and follow-up are shown in Table 4.

### 3.4. Survival Analysis

Median OS was 27 months (95% CI = 21.9, 32.1, n = 258), and OS rates for 1, 3, and 5 years were 82%, 40%, and 22%, respectively (Figure 2A). Median DFS was 13 months (95% CI = 11.0, 15.0, n = 259) and rates for 1, 3, and 5 years were 52%, 15%, and 5%, respectively (Figure 2B). Median survival was significantly (*p* < 0.001) improved in patients with the epithelioid subtype (31 months, 95% CI = 25.9, 36.1) compared to the non-epithelioid (14 months, 95% CI = 10.8, 17.2) subtype (Figure 3A). The epithelioid subtype also had a positive impact on DFS (15 months (95% CI = 13.1, 16.9) vs. 8 months (95% CI = 7.2, 8.8); *p* < 0.001; Figure 3B). Patients with MCR had significantly longer OS (31 months (95% CI = 26.1, 35.9) vs. 15 months (95% CI = 13.1, 16.9); *p* < 0.001) and DFS (14 months (95% CI = 12.1, 15.9) vs. 8 months (95% CI = 5.8, 10.2); *p* = 0.008) compared to patients with an R2 resection status.

In addition, the addition of chemotherapy significantly improved the median OS (32 months (95% CI = 26.8, 37.2) vs. 21 months (95% CI = 13.9, 24.1); *p* = 0.020; Figure 3C), as well as DFS (15 months (95% CI = 12.6, 17.4) vs. 9 months (95% CI = 7.3, 10.7); *p* < 0.001; Figure 3D). UICC stage I vs. ≥II had a significant impact on OS (32 months (95% CI = 27.4, 36.6) vs. 19 months (95% CI = 13.9, 24.2); *p* = 0.028) but not on DFS (14 months (95% CI = 10.5, 17.5) vs. 11 months (95% CI = 9.2, 12.8); *p* = 0.272). The cisplatin dose (high versus low dose) had no significant impact on OS (39 months (95% CI = 16.1, 61.9) vs. 25 months (95% CI = 18.9, 31.2); *p* = 0.103) or DFS (11 months (95% CI = 8.8, 13.2) vs. 13 months (95% CI = 10.9, 15.1); *p* = 0.703).

Using Cox regression for the multivariable analysis (Table 5), the resection status with MCR and histologic subtype with epithelioid histology retained their significant impacts on OS (*p*-values < 0.001), as well as DFS (*p* = 0.010 and *p* < 0.001). In addition, additive chemotherapy still had a significant impact on DFS (*p* = 0.003).

## 4. Discussion

This multicenter study confirmed that patients with operable PM can benefit from CRS and HITOC as part of a multimodal therapy concept regarding disease-free survival and overall survival. In particular, a macroscopic complete resection status and epithelioid subtype were identified as relevant prognostic factors. Median survival of up to 31 months was demonstrated in these patients. The longest OS (median survival of 39 months) was observed in patients receiving high-dose cisplatin at 175 mg/m^2^ BSA, but this parameter did not reach statistical significance compared to low-dose cisplatin. The addition of chemotherapy also appears to have a positive impact on DFS. Thus, multimodal therapy including CRS in combination with HITOC and additional chemotherapy offers an opportunity to significantly improve the survival of selected patients with PM.

Systemic treatment with immunotherapy has become the standard of care for PM patients with locoregional disease that infiltrates surrounding structures [3]. However, immunotherapy has not shown a breakthrough in improving OS compared to chemotherapy specifically for the subset of epithelioid PM [3]. Therefore, a multimodal therapeutic approach is warranted for patients with early-stage disease to improve overall survival. Complete tumor resection and disease control should be accompanied by a good postoperative quality of life for the mostly elderly patients [6,15,17]. Although Canadian colleagues achieved attractive survival rates with a trimodal concept of radiotherapy, EPP and adjuvant chemotherapy in cases of nodal involvement, this concept showed a high morbidity with grade 3 and 4 complications in almost 50% of the cases [18]. In recent years, lung-sparing CRS in combination with HITOC has become a promising treatment concept for localized PM [7,16,18,19]. The preservation of the lung parenchyma provides the patient with a good cardiopulmonary reserve, which is particularly relevant for further systemic therapies within the multimodal therapy concept, as well as for the treatment of tumor recurrence [15,17,20,21]. Therefore, CRS has become the standard surgical technique in many thoracic surgical centers [7,9,11,16,22]. However, the ideal multimodal treatment approach has not yet been identified. In addition, the effects of additive chemotherapy together with CRS and HITOC need to be further investigated [23].

As a relevant outcome parameter for survival, the completion of multimodal therapy has a significant impact on DFS and OS [9,10]. Perioperative morbidity, as well as side effects of neoadjuvant systemic treatment, strongly influence further therapeutic management and the completion of multimodal treatment. In recent years, a multicenter, randomized phase II trial (EORTC 1205) investigated the feasibility of PD or ePD with neoadjuvant or adjuvant chemotherapy [24]. The primary endpoint was successful completion of multimodal treatment, followed by endpoints related to surgical quality, morbidity, and mortality. Initial results from this trial are expected at the end of the year. Our analysis did not show a difference in neoadjuvant or adjuvant systemic treatment. This may be due to the small number of patients receiving neoadjuvant therapy in our cohort.

CRS showed characteristic postoperative morbidity due to the large wound surface of the parietal and visceral pleura. In our study, surgical intervention was required in 14.2% of patients due to prolonged air leak, coagulothorax or cylothorax. In-hospital mortality was 3.5%. As other studies have shown, morbidity and mortality are comparable for this extensive cytoreductive procedure [11,19]. In addition, PD is a defensible cytoreductive surgical procedure within the multimodal therapy of PM compared to known studies including EPP in the multimodal approach [25,26,27]. Although no randomized trials can be performed to evaluate the benefit of HITOC, at least this additional procedure does not show a higher intraoperative complication rate. Our study population has comparable perioperative morbidity and mortality to other centers without HITOC [4,11,19]. Therefore, in our opinion, HITOC can be used as an additional procedure in the fight against PM [7,28].

Based on our retrospective data analysis, the interval from the end of multimodal treatment to tumor recurrence or progression was significantly improved, even after the multivariable analysis, especially in patients receiving additive chemotherapy. It is possible that adjuvant chemotherapy, usually administered postoperatively, can reach residual tumor cells through the large, well-vascularized wound surface. The results of the EORTC trial may provide recommendations on the timing of multimodal therapy. Nevertheless, our results clearly support the inclusion of additive chemotherapy in a multimodal treatment approach. Some centers in our multicenter analysis have not included additive chemotherapy in their standardized treatment regimens in recent years. As a result, we may have reduced the selection bias and the impact of a postoperative decreased general condition on the completion rate of multimodal treatment with additive chemotherapy. This may support our conclusion that additive chemotherapy has a positive impact on disease progression. In accordance with the latest recommendations of the WHO 2021 classification, future analyses may show whether there is an impact on therapeutic decisions and the treatment response when epithelioid mesothelioma is classified as a low-grade or high-grade tumor subtype [29].

The study has some relevant limitations due to its retrospective nature and the analysis of PM patients treated at different institutions over a long period of time. The described cohorts also include the clinical data during the implementation of CRS and HITOC in the different institutions. Based on this, the decision making and the definition of selection criteria for surgical candidates may have influenced the data on perioperative morbidity, MCR, DFS, and OS. Especially due to the recent discussions about multimodal therapy including surgery in the MARS-2 trial, critical patient selection is essential for the best possible survival of the individual patient. However, in summary, carefully selected PM patients could be treated with CRS, HITOC, and additive chemotherapy with low morbidity and promising survival.

Based on the significant impact of additive chemotherapy within this multimodal treatment approach, the NICITA trial (clinicaltrials.gov: NCT04177953), a multicenter trial in Germany of adjuvant chemoimmunotherapy versus chemotherapy alone after CRS for epithelioid PM, will hopefully provide promising results on the impact of immunotherapy within the multimodal treatment of the epithelioid subtype after CRS [30]. In addition, a detailed analysis of PM tumor tissue may help to further understand the tumor biology and develop effective treatment strategies.

## 5. Conclusions

Cytoreductive surgery and HITOC within a multimodal treatment approach have a positive impact on the survival of patients with PM, especially the epithelioid tumor subtype and after macroscopic complete tumor resection. Furthermore, the addition of chemotherapy significantly affects the interval to tumor recurrence compared to CRS and HITOC alone.

## Figures and Tables

**Figure 1 cancers-16-01587-f001:**
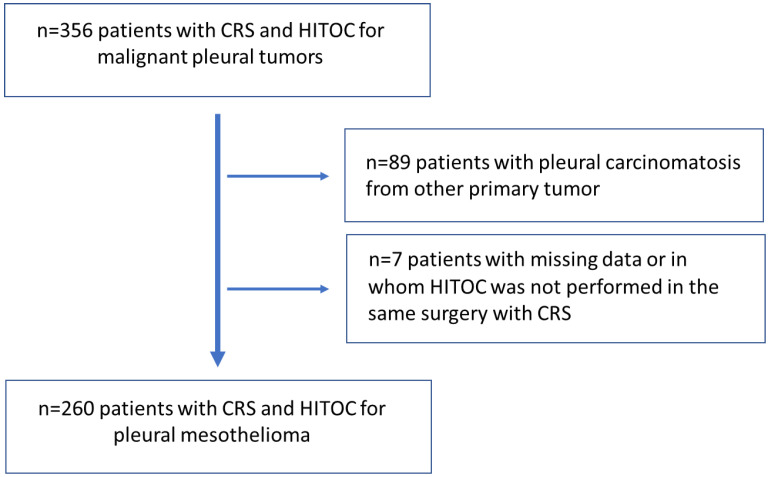
CONSORT diagram of the identification and selection of eligible patients treated with CRS and HITOC for pleural mesothelioma.

**Figure 2 cancers-16-01587-f002:**
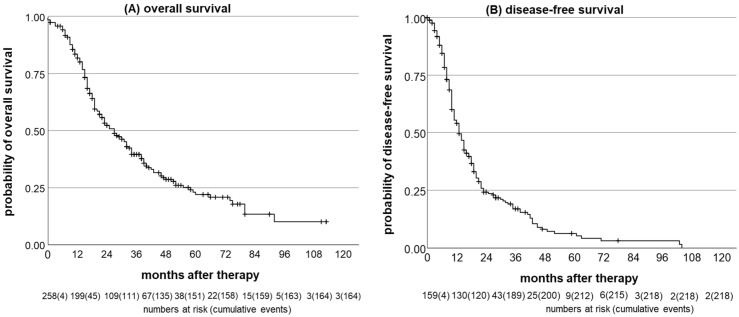
Overall and disease-free survival. (**A**) OS and (**B**) DFS for the whole study cohort of PM patients.

**Figure 3 cancers-16-01587-f003:**
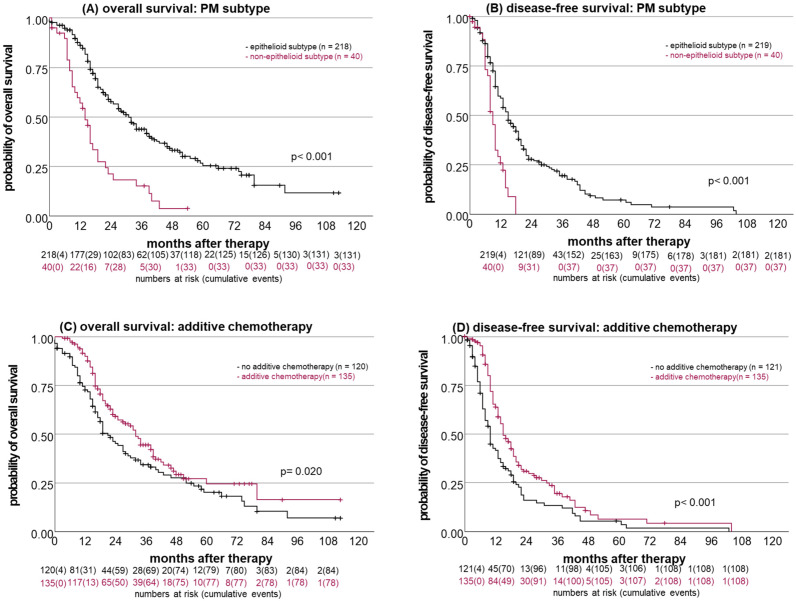
Impacts of the histologic subtype and additive chemotherapy on disease-free and overall survival. (**A**) OS and (**B**) DFS distributed by histologic tumor subtype (epithelioid versus non-epithelioid). (**C**) OS and (**D**) DFS distributed by additive chemotherapy versus no additive chemotherapy.

**Table 1 cancers-16-01587-t001:** Patient characteristics.

Clinical Parameters		n = 260
Sex (n, %)		
	female	49 (18.8)
	male	211 (81.2)
Age (years) (mean ± SD)		65.5 ± 9.0
BMI (m^2^/kg) (mean ± SD)		26.3 ± 3.9
ECOG (n, %)		
	0	159 (61.2)
	1	89 (34.2)
	2	1 (0.4)
	missing	11 (4.2)
Asbestos exposure (n, %)		
	no	56 (21.5)
	yes	171 (65.8)
	missing/unknown	33 (12.7)
Side (n, %)		
	right	156 (60.0)
	left	104 (40.0)
MPM histology (n, %)		
	epithelioid	220 (84.6)
	biphasic	35 (13.5)
	sarcomatoid	5 (1.9)
MPM tumor stage (UICC) (n, %)		
	IA	15 (5.8)
	IB	110 (42.3)
	II	39 (15.0)
	III	90 (34.6)
	IV	6 (2.3)

Clinical data of 260 patients with PM, including histopathological results. BMI—body mass index.

**Table 2 cancers-16-01587-t002:** Surgical approach.

Intraoperative Parameters		n = 260
Resection method (n, %)		
	P/D	60 (23.1)
	eP/D	193 (74.2)
	EPP	7 (2.7)
Resection of diaphragm (n, %)		
	no	87 (54.6)
	yes	173 (66.5)
Replacement of diaphragm (n, %)		
	no	193 (74.2)
	yes	67 (25.8)
Resection of pericardium (n, %)		
	no	142 (54.6)
	yes	118 (45.4)
Replacement of pericardium (n, %)		
	no	199 (76.5)
	yes	61 (23.5)
Resection of chest wall (n, %)		
	no	235 (90.4)
	yes	25 (9.6)
Resection status		
	MCR	222 (85.4)
	R2	38 (14.6)
Blood loss (median, IQR) *		800 (500, 1200)
Transfusion (n, %)		
	no	112 (43.1)
	yes	145 (55.8)
	missing	3 (1.2)
Chemotherapy substances (n, %)		
	cisplatin alone	152 (58.5)
	cisplatin plus doxorubicin	108 (41.5)
Cisplatin dosage (n, %)		
	low dose (≤125 mg/m^2^ BSA)	191 (73.2)
	high dose (>125 mg/m^2^ BSA)	70 (26.8)
Intraoperative complication		
	no	252 (96.9)
	yes	8 (3.1)

* Blood loss was documented in 200 out of 260 (76.9%) patients. The number of transfusions was computed for 257 of 260 (98.8%) patients. Detailed data on the extent of resection during CRS and HITOC.

**Table 3 cancers-16-01587-t003:** Postoperative data.

Postoperative Parameters		n = 260
Postoperative complications (n, %)		
	no	118 (45.4)
	yes/Clavien–Dindo classification	142 (54.6)
	I	12 (4.6)
	II	52 (20.0)
	IIIa	28 (10.8)
	IIIb	37 (14.2)
	IVa	1 (0.4)
	IVb	3 (1.2)
	V	9 (3.5)
Revision (n, %)		
	no	224 (86.2)
	yes/revision due to	36 (13.8)
	coagulothorax	9 (3.5)
	pleural empyema	3 (1.2)
	chylothorax	5 (1.9)
	parenchymal fistula/bronchial fistula	8 (3.1)
	other	11 (4.2)
Prolonged ventilation > 24 h (n, %)		
	no	251 (96.5)
	yes	9 (3.5)
Extubation in the OR (n, %)		
	no	21 (8.1)
	yes	239 (91.9)
ICU stay [days] (median, IQR)		2 (1, 4)
Hospital stay [days] (median, IQR)		20 (15, 29)
In-hospital mortality (n, %)		9 (3.5)

Overview on postoperative complications and surgical revisions together with data on hospitalization. ICU—intensive care unit, OR—operating room.

**Table 4 cancers-16-01587-t004:** Multimodal treatment and follow-up.

Postoperative Course		n = 260
Neoadjuvant chemotherapy (n, %)		
	no	197 (75.8)
	yes	63 (24.2)
Adjuvant chemotherapy (n, %)		
	no	171 (65.8)
	yes	86 (33.1)
	missing	3 (1.2)
Additive chemotherapy (n, %)		
	no	121 (46.5)
	yes	136 (52.3)
	missing	3 (1.2)
Adjuvant radiotherapy (n, %)		
	no	199 (76.5)
	yes	58 (22.3)
	missing	3 (1.2)
Recurrence (MCR, n = 222) (n, %)		
	no	52 (23.4)
	yes	170 (76.6)
Progression (R2, n = 38) (n, %)		
	no	13 (34.2)
	yes	38 (65.8)
Location of recurrence/progression (n = 195) (n, %)		
	locoregional	157 (80.5)
	distant metastasis	15 (5.8)
	locoregional and distant metastasis	17 (8.7)
	missing	6 (3.1)
Site of recurrence/progression (n = 195) (n, %)		
	ipsilateral	174 (89.2)
	contralateral	6 (3.1)
	missing	15 (5.8)
Therapy for recurrence/progression (n = 195) (n, %)		
	no	11 (5.6)
	unknown	16 (8.2)
	yes/type	168 (86.2)
	surgery	19 (9.7)
	chemotherapy	117 (60.0)
	radiotherapy	30 (15.4)
	supportive care	14 (7.2)
	other	3 (1.5)
Time between the start of primary therapy until the end of follow-up in months * (reverse Kaplan–Meier OS) (median, 95% CI)		48 (38, 58)

* A total of 258 of 260 (99.2%) patients; two patients were excluded from analyses as either the date of start of primary therapy or the date of death was missing. Both patients died. Additive treatment within a multimodal treatment concept including follow-up data.

**Table 5 cancers-16-01587-t005:** Cox regression analysis of OS and DFS.

	Overall Survival				Disease-Free Survival			
	Sig.	HR	95% CI		Sig.	HR	95% CI	
cisplatin dosage	0.350	0.81	0.53	1.25	0.831	1.04	0.74	1.45
chemotherapeutic agent	0.114	1.35	0.93	1.95	0.556	1.10	0.80	1.53
resection status	<0.001	2.25	1.45	3.49	0.010	1.72	1.14	2.60
histological subtype	<0.001	2.27	1.50	3.41	<0.001	2.43	1.63	3.61
UICC stage	0.544	1.11	0.79	1.57	0.986	1.00	0.74	1.34
postoperative AKI	0.897	1.03	0.64	1.65	0.515	1.14	0.77	1.70
additive chemotherapy	0.184	0.79	0.56	1.12	0.003	0.64	0.47	0.85
time between the initial diagnosis and start of primary therapy	0.712	1.01	0.97	1.05	0.772	1.00	0.98	1.04
	N = 255 *; n = 162 dead, n = 93 censored				N = 256 **; n = 218 recurrence/progression or dead, n = 38 censored			

* Five out of 260 (1.9%) patients were excluded due to missing information about additive chemotherapy (n = 3) or time points (n = 2). ** Four out of 260 (1.5%) patients were excluded due to a missing date of the start of primary therapy (n = 1) or missing information about additive chemotherapy (n = 3). For the one patient with a missing date of death, the date of recurrence/progression was available. Thus, the patient was included in recurrence/progression-free survival analysis. Reference categories: low-dose cisplatin, cisplatin alone, resection status R1, epithelioid MPM, UICC stage I, no postoperative acute kidney insufficiency (AKI), and no additive chemotherapy.

## Data Availability

Data are available on request due to privacy and ethical restrictions.

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
