# Peer review of "Multimodal Treatment of Pleural Mesothelioma with Cytoreductive Surgery and Hyperthermic Intrathoracic Chemotherapy: Impact of Additive Chemotherapy"

_cancers, 2024, doi:10.3390/cancers16081587_

Round 1

Reviewer 1 Report

Comments and Suggestions for Authors

Well done! Few things to take under consideration.

1. The criteria of macroscopic complete resection is rather rare to the routine, so I suggest to be more detailed of how you "sign" the complete resection.

2. The limitations of the study are there, write something about the low-high grade mesotheliomas and the histological reports(WHO 2021).

3. It would be a point in favor if you could find the cases of certain exposure to asbestos (and maybe the type of asbestos fibers, since the have different biological behavior in pathology).

Author Response

Dear Reviewer,

thank you very much for your time to critically review our manuscript.

Please find our responses to your comments below.

1. The criteria of macroscopic complete resection is rather rare to the routine, so I suggest to be more detailed of how you "sign" the complete resection.
We added the defintion and associated literature to the text to further characterize macroscopic complete resection.

2. The limitations of the study are there, write something about the low-high grade mesotheliomas and the histological reports(WHO 2021).
Thank you very much for adding this intersting point! We added information on this issue to the text together with relevant literature for the reader.

3. It would be a point in favor if you could find the cases of certain exposure to asbestos (and maybe the type of asbestos fibers, since the have different biological behavior in pathology).

According to the information provided in Table 1, 171 patients had former asbestos exposure. Due to the retrospective character of the study, asbestos status remains unknown in 33 cases. The type of asbestos fibers is not characterized in details in our cohort. This is a great suggestion for future analysis.

Reviewer 2 Report

Comments and Suggestions for Authors

The present manuscript “Multimodal Treatment of Pleural Mesothelioma with Cytoreductive Surgery and Hyperthermic Intrathoracic Chemotherapy: Impact of Additive Chemotherapy” by Klotz et al. is indeed interesting. Some minor points needs to be addressed. Please find my issues below.

·         Please provide a CONSORT diagram describing the study in- and exclusion

·         Please also give number of adverse events/perioperative death in the abstract since HITOC was related to significant morbidity and mortality in the early days

·         How many pts received immune therapy?

·         Please add p values and number at risk to the survival graphs

Author Response

Dear Reviewer,

thank you very much for taking the time to critically review our work!

Please find our detailed responses to your comments below.

1. Please provide a CONSORT diagram describing the study in- and exclusion

We created a CONSORT diagram to visualize the in- and exclusion criteria.

2. Please also give number of adverse events/perioperative death in the abstract since HITOC was related to significant morbidity and mortality in the early days

We included the number into the abstract.

3. How many pts received immune therapy?

Since patient recruitment was conducted between 2008 and 2019, patients did not receive immune therapy due to missing approval for mesothelioma in this time period.

4. Please add p values and number at risk to the survival graphs

We rearranged the survival graphs and added p-values and numbers at risk.